# Energy Consumption of Beverage-Bottling Machines



**Isabel Anna Osterroth [1],\* and Tobias Voigt [2]**

1   TUM School of Life Sciences, Technical University of Munich, Alte Akademie 8, 85354 Freising, Germany
2   Brewing and Beverage Technology, Technical University of Munich, Weihenstephaner Steig 22, 85354 Freising, Germany; tobias.voigt@tum.de
\*   Correspondence: isabel.osterroth@tum.de

**Abstract:** Sustainability is a megatrend influencing the beverage industry. Knowledge of the consumption behavior and suitable metrics are required for energy optimization strategies. Machine efficiency and energy consumption are intermixed in common parameters, e.g., customary specifications refer to the energy consumption for a specific number of products (e.g., kWh/1000 fillings). This does not reflect the influence that inevitable breakdown times have on the energy consumption (e.g., malfunction, lack, and tailback situations within the material flow). While specific energy performance indicators are useful as a benchmark, it does not provide reliable information to verify plant specifications, or to have a source-related cost allocation as a basis for a weak point analysis. In this work, energy and operational data were analyzed, in order to find a generic description of the operational-state related consumption behavior. Therefore, empirical data on the effective electrical energy and operational state data were collected on machine level of two representative bottling plants and for additional single machines. In the frequency distributions of the discrete values of the measured electrical energy data, three main peaks were found. These can be correlated to operational states such as state-related energy demand level. The change from one demand level to another was found to be reproducible.

**Keywords:** energy consumption; bottling; energy performance indicators

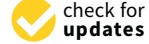

## 1. Introduction

Packaging is the last step in food and beverage production and has an important influence on the product quality, shelf life, and marketing aspects. The packaging of beverages and liquid food is performed by a complex network of machines, which are interlinked with buffering transport units for primary or secondary packages (e.g., bottles and pallets) [1]. The packaging process comprises unpacking and cleaning of the packages, filling them with the product and sealing of the package, examination of the packed product, and packaging in a secondary package or transport package. The legal developments, economic, social, and technical trends, and rising energy costs have shifted the focus of industry to energy. In particular, companies working in the field of the cost-driven beverage and food industry increasingly analyze the energy and media consumption and are looking for optimization strategies. Olajire found, that despite significant technological improvements over the last years, energy consumption, water consumption, and water usage and waste remain major environmental challenges in the brewing industry [2]. The cost of energy is already an important factor in a company's cost structure in the food and beverage industry, which is why monetary considerations have an influence on this development. Recent social and political drivers (e.g., the UN Sustainable Development Goals [3]) accelerate the demand for a more energy efficient production process. The leading worldwide breweries and beverage companies have recently published ambitious targets to become carbon neutral in the nearer future (see e.g., sustainability goals of Anheuser-Busch InBev with 25% reduction in $CO_2$ emissions across the value chain until 2025 [4], Heineken with 0% $CO_2$ emission until 2030 [5], and Coca Cola with net-zero carbon emissions by 2050 [6]). Suitable tools and metrics are required to fulfill these targets.

*1.1. Beverage-Packaging Technology: Energy Consumption of Bottling Machines*

Beverages and liquid foods are packaged in industrial bottling plants consisting of an arbitrary number of cooperating machines and aggregates (manufacturing stations), which depend on the product and packaging type. These are all interlinked with buffering transport elements (mainly conveyors). Various containers, such as bottles (disposable and returnable bottles, PET bottles, and glass bottles), cans, barrels, or kegs, are used for packaging and for transporting different beverages (beer, wine, alcohol-free beverages, etc.). Owing to the modular design of the systems, the different requirements of beverage filling can be met by a variety of machines. According to Manger [7], the components of a beverage filling line are essentially the same:

- Machines unloading and loading pallets, as well as unpacking the reusable containers made of glass and plastic from the transport boxes (e.g., unloading machines, unpacking machines, and new glass slippers);
- Machines ensuring the cleaning and control of the outer packaging (e.g., crate washing machines);
- Machines ensuring the cleaning and inspection of the packaging material (returnable bottles) (e.g., bottle and empty bottle inspectors, bottle-cleaning machines);
- Machines filling and sealing the packaging (e.g., bottle filling machines with a combined capper);
- Machines labeling and checking the packing materials (e.g., labeling machines);
- Packing machines (e.g., shrink packers, palletizers);
- Conveyors for containers, packs, and pallets (e.g., chain conveyors with a drive engine).
- In addition, the following are used when needed:
- Machines for bottle manufacturing (e.g., stretch-blow molders for PET bottles);
- Machines improving the biological shelf life (e.g., pasteurizers);
- Machines for cleaning and disinfection (e.g., CIP systems);
- Sorting systems of returnable containers (e.g., selective unpacking machines);
- Machines for producing soft drinks (e.g., beverage mixing and carbonizing plants).

The requirements for individual plant components are related not only to the container type and size and the packaging material (glass, PET, aluminum, or steel), but also to the beverage properties (presence of $CO_2$ and viscosity). During the filling and packaging of the beverages and liquid foods, different media are consumed by different processes. The typical consumption of beverage-bottling, which has a significant commercial effect, includes the following:

- Electrical power and compressed air;
- Thermal power;
- Media such as cold water, lye, additives, and lubricants.

In order to operate a bottling plant, all bottling and packaging machines require electrical power, in addition to cleaning machines (bottle-cleaning machines and crate washers), and pasteurizers require thermal energy (see Table 1). In addition, some machines use compressed air, usually created by electrically driven compressors (mainly packers and palletizers). Water is mainly necessary for the cleaning machines of primary and secondary packaging materials (bottles, crates) and during the pasteurization process.

Electrical power is usually used as the main drive for machines moving containers, drive pumps, to perform their basic mechanical functions as well as for the production of compressed air. The consumption of electrical energy is usually linked directly to the function performed. Any interruption of the function or reduction of the output power also leads to reduced consumption in electrically driven motors and pumps. Thermal energy is usually used to heat various media (e.g., heating of products, heating of the lye baths in bottle-cleaning machines, and supply of hot water). The consumption of thermal energy by a machine can be decoupled from its operational state since the media might be heated even in the case of a malfunction in the machine. Compressed air is used during beverage filling either as an energy carrier (stretch-blow molding of PET bottles, transport of empty PET

bottles), for signal transmission (pusher), or for cleaning (blow-off, drying). Compressed air is a comparatively energy-intensive energy carrier and is produced using electrical power. Freshwater is usually used to clean machines and is also used in bottle-cleaning machines (cleaning of returnable containers). Other media, such as lye, belt lubricants, and additives, are used mainly in bottle-cleaning machines and during the transportation of containers.

**Table 1.** Energy and media consumption of bottling machines according to the minimum and order related specifications of beverage filling lines, own table according to [8].

| | Electrical Power | Compressed Air | Heat | Water |
|---|:---:|:---:|:---:|:---:|
| Stretch-blow molder | × | × | | |
| Depalletizer | × | × | | |
| Unpacker | × | × | | |
| Crate washer | × | | × | × |
| Bottle-cleaning machine | × | | × | × |
| Labeling machine | × | × | | |
| Pasteurizer | × | | × | × |
| Packing machine/tray packer | × | × | | |
| Palletizer | × | × | | |

Over the last years, mainly industry associations and consulting companies have published consumption values and benchmarks. The data are predominantly related to the total production processes or single-process units (e.g., packaging). Most of the data are published as total consumption for the production or differentiated into thermal and electrical energy, which is mainly related to the product volume (e.g., 1 hl of sales beer). The cost of supplying energy to a brewery differs worldwide from 3% to 10% of the total budget [9–11]. Over the last 10 years, the required volume of water to produce 1 hl of beer decreased from 5.0–5.2 hl to 4.2–4.3 hl [9,12–14]. Moreover, 16.5–30% of the total heat and 12–35% of the total electrical power are needed for the bottling process [15–17]. Hauser and Shellhammer analyzed the sustainability challenges in beer production and found that packaging has a large share on the environmental impact of beer production [18]. In contrast to breweries, only a limited amount of consumption data has been published for manufacturers of nonalcoholic beverages. No detailed machine-based consumption data and analyses of beverage-packaging machines have been published in the scientific literature, with the exception of one report [19] in the early 1980s. No energy data correlated to the operational state of single machines have been published either.

For in-depth analyses and optimization measures, a detailed consideration of the energy consumption on the machine level is lacking. Measurements on bottling and packaging machines in industrial applications related to this research indicated that there is a significant difference between the installed load and the average measured consumption, which indicates a saving potential. The influence of the machine efficiency on the energy consumption is not considered in commonly used key performance indicators. This does not consider, e.g., the influence of breakdown times, which, because of equipment failure, lack, and tailback situations, inevitably occur during the production. While considering the energy consumption of a defined amount of product is useful for the comparison of the performance of two plants and management decisions, it does not provide reliable information to verify specified machine-based consumption data or for a source-related cost allocation as a basis for a weak point analysis and machine, process, and automation optimization. Some manufacturers have started to specify a consumption level related to the nominal speed. However, consumption values for planned or unplanned stops of machines were missing in the past. With *VDMA 8751:2019-03 Packaging machinery (incl. filling machinery)—Specification and measurement of energy and utility consumption* [20], recently, a normative directive was published to close this gap.

Osterroth et al. have published a summary of bottling related energy KPIs and a survey for the German beverage bottling industry. It was found in this publication that

the available and published data are not yet detailed enough for a modeling approach [21] that could be used as a tool for energy optimization of bottling plants. The available data are not suitable for any comparison, as the survey approaches and system boundaries vary greatly. No correlation between the current machine or process state and the energy consumption has been investigated, and considerations to reduce the energy and media consumption have been limited locally to individual system components or to a high-level view so far. With detailed knowledge regarding the consumption behavior and with a generic model mapping this behavior, the main consumers can be detected, identifying optimization potential in the production process, the machines, and the plant automation.

### 1.2. Models Describing the Energy Consumption in Industrial Manufacturing Lines

Some models that describe the energy consumption of machines in manufacturing lines can be found in the scientific literature: Dietmair and Verl published a generic energy consumption model for decision making and energy efficiency optimization in manufacturing for plants, machines, and components based on a statistical discrete event formulation [22]. Lees et al. described a utility consumption model for real-time load identification in a brewery [23]. Several fuzzy and neurofuzzy models have been developed for electrical load forecasting (e.g., [24,25]). One example of the use of mathematic modeling is the energy optimization of refrigeration systems in breweries. Xu et al. developed a case-based reasoning (CBR) energy consumption model for cutting periods in CNC lathes [26]. Cataldo et al. worked on the modeling, simulation, and evaluation of energy consumption for a manufacturing production line [27]. For tooling machines, a calculation method was developed by Kuhrke using "energy blocks", which are consumption patterns based on functional changes in the machine operations, comparable to operational states. For those machine tools, for mass production, a state-based model was defined, describing the detailed single phases of the production process, based on energy blocks [28]. This model was published in VDMA 34179 as a measurement instruction determining the energy and resource demand of machine tools for mass production [29] and was updated in 2019 [30]. Braun et al. discussed a state-based energy consumption modeling approach for the example of wireless sensors and wireless local area networks, considering the consumption patterns for single states and state transitions [31].

Owing to their high interlinking level and high output (up to 120,000 packages per hour), packaging and bottling machines differ from other manufacturing plants. It was found that state models from other industries cannot be transferred to bottling plants directly. Customary specifications refer to the energy consumption for a specific number of products (e.g., kWh/1000 fillings). Osterroth et al. published a state related simulation model for bottling plants to close this scientific gap [32], which is completed by the here-presented data analysis. As part of this work, the energy behavior of food and beverage packaging machines were analyzed systematically, considering the recent operational state of the machines for the first time.

### 1.3. Purpose

The data presented in this paper were unpublished as of now, extend the existing non-sufficient database, complete the normative direction VDMA 8751:2019 and the published simulation approach, and give additional insights in the state-related energy consumption behavior for a larger number of considered machines. It is assumed that a detailed generic model can be used as the basis for the in-depth analysis of the consumption behavior for identifying inefficient production times and for conducting further research on the modeling and simulation of the energy consumption of food and packaging plants. The purpose of this study is to gain fundamental knowledge regarding the operational state related consumption behavior of packaging and bottling machines based on detailed empirical data from industrial plants. Detailed empirical electrical energy data on machine level shall be analyzed to identify main consumer and to provide generic statements for future modeling and forecasting approaches and optimization measures.

## 2. Data Acquisition

Owing to the lack of reliable data, electrical energy consumption data and operational state data of 20 machines in industrial applications were considered to characterize the consumption behavior of bottling and packaging machines in the food and beverage industry.

### 2.1. Acquisition of Energy Data

The electrical power consumption was recorded with active power meters of the highest available sample rate (1 or 2 s) as effective electrical power (kW). Depending on the technical possibilities, integrated (Janitza UMG 96RM-PN, measurement accuracy: $\pm0.2\%$) or mobile measuring instruments (Fluke 435 II, measurement accuracy: $\pm1\%$) were used. All active power meters were state of the art and calibrated. The measured discrete values were recorded in a MS SQL database with a timestamp.

### 2.2. Acquisition of Operational State Data

Physical machine state changes of production machines and lines can be described by established state models mainly used for machine control and automatization. Beverage bottling plants and packaging machines are generally described by some state models in scientific research and industrial application, as discussed below.

The American National Standards Institute (ANSI) and the Instrumentation, Systems and Automation Society (ISA) published the international standard ANSI/ISA-S 88 [33], addressing batch process control in which the models, terminology modes, and states of single process units or machines are defined. These definitions are not part of the standard but are generally defined. The model describes operating modes (automatic, semi-automatic, and manual) and operating conditions (idle, running, complete, paused, holding/held, restating, stopping/stopped, aborting/aborted). In the operating state model, a focus is placed on the transitions between the final states. For the application to packaging machines, a technical report is published [34].

The "Weihenstephan Standard" defines a communication protocol, data tags to acquire relevant production data and procedures for efficiency analysis (e.g., OEE indicators, cost transparency), tracking and tracing of batches, transport units and orders, and the monitoring of production processes and the product quality. It includes a state model, describing the physical machine states with its operating mode (off, automatic, semi-automatic, manual), its program (e.g., production, start up, clean), and its operational state [35]. The "Weihenstephan Standard" state definitions are harmonized with the definitions of the OMAC Packaging Machine Language Working Group, which has also published a uniform state model to describe the different operation states and user actions with packaging machines [36].

For the analysis of the production and performance of packaging and beverage bottling lines a time model is established in accordance with DIN 8743 [37] The machine working time $t_m$ is defined as the theoretically available time (24 h/7 days a week). Deducting the idle time (e.g., weekend, holidays), the machine work time is defined. The operating time is defined as the machine working time reduced by scheduled downtimes (e.g., cleaning/maintenance). During the machine working time breakdowns will occur due to internal and external failures. The machine running time is the working time reduced by the breakdown time. The time model is correlated with the physical machine (see Figure 1).

The state data were collected time discrete every second or every two seconds (as available) by the manufacturing execution systems (MES) in accordance with the industrial "Weihenstephan Standard" based on the time model according to DIN 8743 [37]. If no MES was available, the data were collected by the Weihenstephan Test Tool, collecting the state information directly on the PLC of the machine or manually (handwritten notes). The state data were recorded event discrete with a start time and an end time of the state. The state information is based on a state model and indicates whether a machine is producing (operating state "operating") or waiting to produce in a suspended state due to an internal

cause ("failure", "held", "emergency stop") or external cause ("lack", "tailback", "idle", "prepared"). There was no data available for the machine mode (on/off) and program (e.g., "Production", "Maintenance", "Cleaning"). The state information can be assigned to time intervals in accordance with the time model of DIN 8743 [37]. Before analyzing the state data, the correct acquisition of the machine states was verified at least three times for each state by checking the state information in the recording database in parallel with the actual machine behavior. The data acquisition system does not record information about the states of the bottle, crate and pallet conveying systems, and the inspection machines. It is for this reason that they were not considered in this work.

| theoretical available time (24 hours, 7 days a week) $t_T$ | | | | | | |
|---|---|---|---|---|---|---|
| machine working time $t_W$ | | | | | | idle time |
| operating time $t_O$ | | | | | scheduled down time $t_D$ | ▪ idle |
| running time $t_R$ | | | unplaned down time $t_F$ | | | ▪ undefined |
| quality time $t_Q$ | scrape time $t_{LQ}$ | performance loss time $t_{LP}$ | ▪ failures ▪ lack ▪ tailback | | ▪ cleaning ▪ chanceover ▪ maintenance ▪ repair | |

**Figure 1.** Time model according to DIN 8743 [37] and example states (grey).

### 2.3. Considered Bottling Plants and Data

To obtain representative data, five bottling plants using typical packaging types (returnable glass and PET bottles) and products (water, soft drinks, and beer) were considered for the data analysis. All machines of two bottling plants were analyzed, and for the other ones, only the main consumers (bottle cleaning machine or stretch-blow moulder) were taken into account. Additionally, a packaging machine for tray packs was analyzed having a control machine from a different packaging application. The following Table 2 shows a summary of the considered machines.

**Table 2.** Summary of data acquisition on the considered machines.

| | Electrical Consumption Data | Sample Rate | Number of Machines | Packaging Type | Product |
|---|---|---|---|---|---|
| Bottling plant 1 | active power meter, integrated | 2 s | 10 | returnable glass bottle | water, soft drinks |
| Bottling plant 2 | active power meter, integrated | 2 s | 9 | PET bottle (single use) | water, soft drinks |
| Bottling plant 3 | active power meter, mobile | 1 s | 1 | returnable glass bottle | beer |
| Bottling plant 4 | active power meter, mobile | 1 s | 1 | returnable glass bottle | water, soft drinks |
| Bottling plant 5 | active power meter, mobile | 10 s | 1 | PET bottle (single use) | water, soft drinks |
| Packaging machine A | active power meter, mobile | 1 s | 1 | tray packs | various |

## 3. Results

### 3.1. Development of Analysis Methods

For the in-depth analysis of the measured energy consumption data of packaging and bottling machines, the frequency distribution of the measured discrete effective electrical power values was plotted as a histogram for every single machine (see Figure 2). The class width of the histogram was defined depending on the number of the single values and the distribution. It was assumed that peaks represent a so-called energetic demand level. As

the height of the peaks is assumed to be the result of the machine use (occurring operation states) during the measurements, the peaks were qualitatively evaluated.

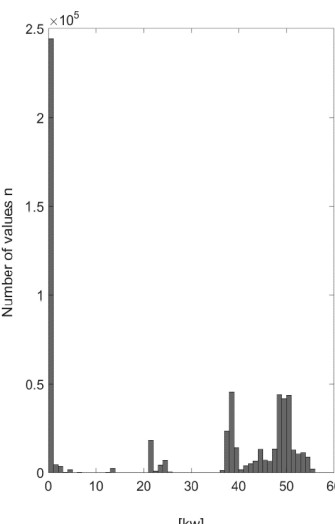

**Figure 2.** Example of frequency distribution of the measured effective power values discrete.

The first analysis results plotting the electrical energy and the occurring operational states indicated a correlation between the consumption and the operational states (see Figure 3).

For the purpose of illustration and analyzing the correlation between the energy consumption and operational states, a diagram was developed showing the measured effective electrical power plotted as a time-discrete 2D line plot with colored event-discrete intervals (areas) in a background layer representing the occurring operational states. The boundaries of the intervals are described by the start time and end time of the operational state ([start_timestate n; end_timestate n], see Figure 4). Therefore, every defined operational state was assigned a color (see Table 3). Similar operational states were summarized to one color, for example, lack and lack in branch line.

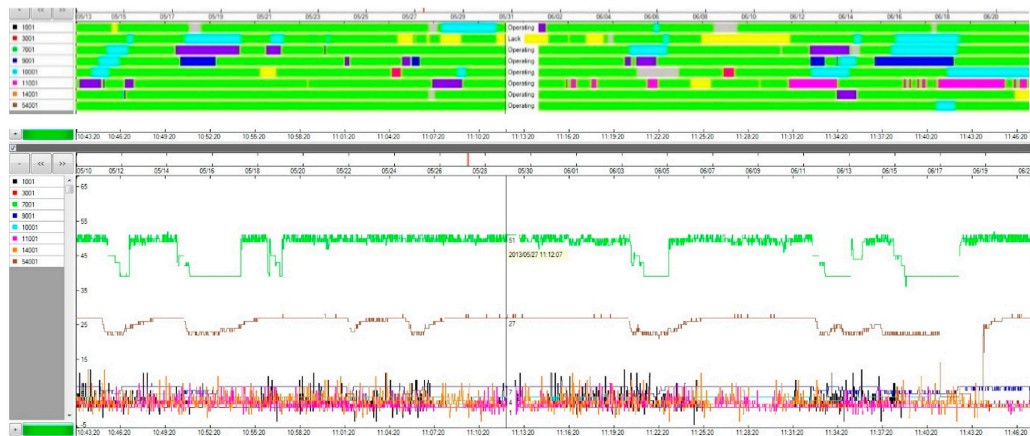

**Figure 3.** Example of the measured effective electrical energy raw data and empirical operational state raw data plotted over time for different machines (Machine ID: 1001, 3001, etc.).

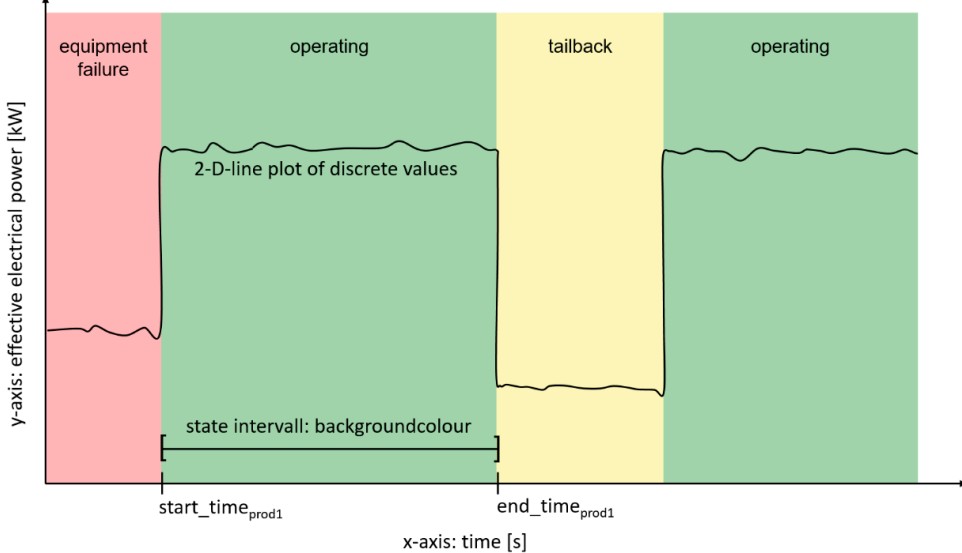

**Figure 4.** Example plot for evaluating the correlation between the operational state and energy consumption.

**Table 3.** Operational states and related background colors in the plot.

| Operational State | Color Coding |
|---|---|
| Prepared | Orange |
| Lack | Blue |
| Tailback | Yellow |
| Lacking branch line | Blue |
| Tailback branch line | Yellow |
| Operating | Green |
| Equipment failure | Red |
| External failure | Dark orange |
| Held | Grey |
| Idle | Grey |

*3.2. Electrical Energy Consumption of Packaging and Bottling Machines*

For the considered bottling plants, the average consumption for single machines was calculated. Owing to the limitations in the available measurement instruments, no data on the conveyor system of the returnable glass bottling plant were available. For the PET bottling plant, the total value of the bottle conveyor and the total value of the pallet transport were available.

3.2.1. Selected Absolute Consumption Values on the Example of a PET Bottling Plant (Plant 2)

Table 4 shows the selected measured electrical energy consumption for all considered machines on the example of the PET bottling plant (Plant 2) for three time periods of 24 h, each during a production phase. The stretch-blow molder was identified as the main consumer with an average consumption of 157 kW, which is more than 50% of the total consumption of the line (245 kW on average). The conveyor systems had an average consumption of 6% of the total consumption (4% for the bottle conveyor and 2.2% for the pallet conveyor).

**Table 4.** Measured electrical energy consumption for the example of a PET bottling plant for three time periods ($t_i$) of 24 h each (sorted by main consumers), average consumption, standard deviation, and percentual share on total consumption.

| | $\sum t_1$ [kWh] | $\sum t_1$ [kWh] | $\sum t_1$ [kWh] | Avg. Consumpt. [kW] | Standard Deviation [kW] | Share [%] |
|---|---|---|---|---|---|---|
| Stretch-blow molder A | 3019.0 | 4281.0 | 3998.6 | 156.9 | 22.5 | 64 |
| Labeler A | 343.6 | 449.7 | 513.2 | 18.1 | 2.9 | 7.4 |
| Shrink packer A | 326.1 | 448.5 | 446.6 | 17.0 | 2.4 | 6.9 |
| Cooler A (stretch-blow molder) | 267.6 | 385.9 | 373.6 | 14.3 | 2.2 | 5.8 |
| Palletizer A | 259.2 | 365.8 | 356.3 | 13.6 | 2.0 | 5.6 |
| Conveyor (bottles) A | 189.2 | 258.8 | 254.4 | 9.8 | 1.3 | 4.0 |
| Filler A | 157.8 | 203.7 | 200.2 | 7.8 | 0.9 | 3.2 |
| Conveyor (pallets) A | 107.0 | 141.9 | 135.5 | 5.3 | 0.6 | 2.2 |
| Handle application A | 46.9 | 62.2 | 61.2 | 2.4 | 0.3 | 1.0 |
| Conveyor belt lubrication A | 1.2 | 1.3 | 1.3 | 0.1 | <1 | <1 |
| Preform feed A (stretch-blow molder) | 1.2 | 1.2 | 1.2 | 0.1 | <1 | <1 |
| Total consumption, $t_i$ (24 h each) | 4718.8 | 6600.0 | 6342.0 | 245.3 | | |

Figure 5 shows a comparison of the installed load and the specified consumption for nominal speed as well as the measured average consumption and the consumption during production times for the example of the selected machines of bottling plant 2. The figure shows a wide deviation between the specified values and the measured values. For the stretch-blow molder and filler block, including the cooler and preform feed, less than 50% of the installed load was measured. Peak loads were not found with this interval of measurement (one value per second).

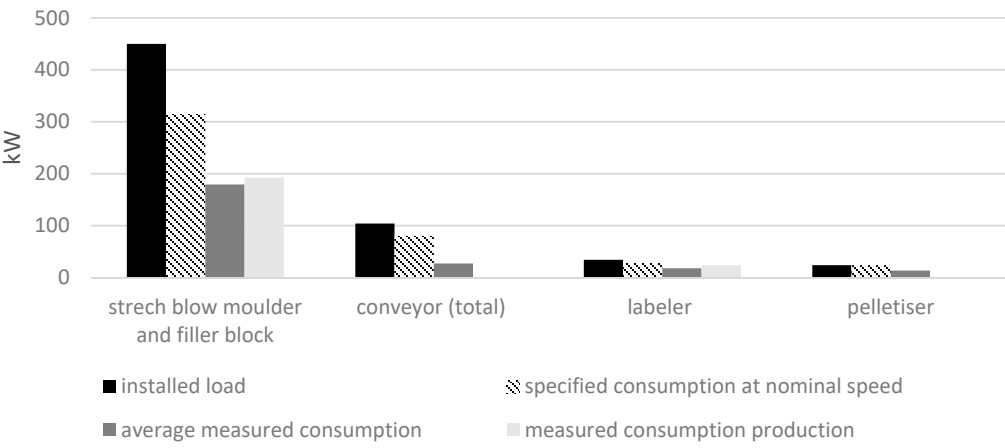

**Figure 5.** Installed load, consumption nominal speed, measured average consumption, and measured consumption during production.

Additionally, the energy consumption during longer unproductive times (e.g., weekends) was analyzed. Table 5 shows a summary of the average measured consumption during a time period of >24 h, as well as the percentage of consumption compared to the average value. While the main consumer, the stretch-blow molder, has a low percentage of consumption during these times, the labeler and the filler have a higher percentage of consumption during these times. For the stretch-blow molder, the filler's and the labeler's detailed operational state data were available.

**Table 5.** Measured average power consumption during an unproductive time period of >24 h.

|  | Average Consumption [W] | Standard Deviation [W] | [%] of Average Consumption |
|---|---|---|---|
| Labeler | 4112 | 2341 | 33 |
| Filler | 2020 | 385.1 | 16 |
| Stretch-blow molder | 1715 | 62.4 | 14 |
| Conveyor (pallets) | 1082 | 25.4 | 9 |
| Conveyor (bottles) | 1058 | 465.7 | 9 |
| Handle application | 1024 | 836.4 | 8 |
| Palletizer | 581 | 2.7 | 5 |
| Handle application | 407 | 3.4 | 3 |
| Cooler (stretch-blow molder) | 231 | 4.8 | 2 |
| Preform feed (stretch-blow molder) | 49 | 1 | <1 |
| Conveyor belt lubrication | 48 | 1.6 | <1 |
| Total consumption | 12,327 |  |  |

Table 6 summarizes the specific energy consumption for those machines for a time period of one production week and specifies the occurring operational states as well as the consumption during these times.

**Table 6.** Specific energy consumption of the main consumer for a time period of one week and an analysis of the consumption related to the operational behavior of the machines. No state data are available for the shrink packer and palletizer.

|  | Stretch-Blow Molder | | Filler | | Labeler | |
|---|---|---|---|---|---|---|
|  | [h] | [Wh/1000 Bottles] | [h] | [Wh/1000 Bottles] | [h] | [Wh/1000 Bottles] |
| Production | 119 | 4290.2 | 117 | 212.8 | 121 | 581.6 |
| Prepared | 4.6 | 12.4 | 5.2 | 8.1 | 3.5 | 3.3 |
| Lack | 0.1 | 1.3 | 0.0 | 0.0 | 4.6 | 4.4 |
| Tailback | 6.8 | 145.8 | 7.8 | 9.9 | 4.8 | 4.6 |
| Equipment failure | 8.1 | 195.6 | 4.9 | 6.5 | 2.5 | 2.2 |
| Held | 29.2 | 48.7 | 33 | 37.4 | 31.7 | 25.2 |
| Total |  | 4694.0 |  | 274.8 |  | 621.2 |
| Availability of the machine |  | 71.0% |  | 69.8% |  | 72.0% |

Figure 6 shows the typical consumption structures for a PET bottling plant (A) for single-use bottles and a glass bottling plant (B) for returnable glass bottles. The main consumers of these types of bottling plants are the stretch-blow molders (PET) and the bottle-cleaning machines (glass).

### 3.2.2. Discussion of the Absolute Consumption Values

Both considered lines have a main consumer, which has a significant share on the total consumption of the total bottling plant (stretch-blow molder: >60% for PET bottles; bottle-cleaning machine: >45%), which should be the focus of optimization considerations.

The installed load and the specified consumption differ from the measured consumption during production times by up to 60%. The common considerations using these values for an estimation of the energy consumption will have a limited accuracy. For optimization purposes and business decision criteria, these values should be verified by measurements of the actual consumption. For modeling and forecasting of the energy consumption, the implemented parameter should be verified at least by single measurements on the considered plant.

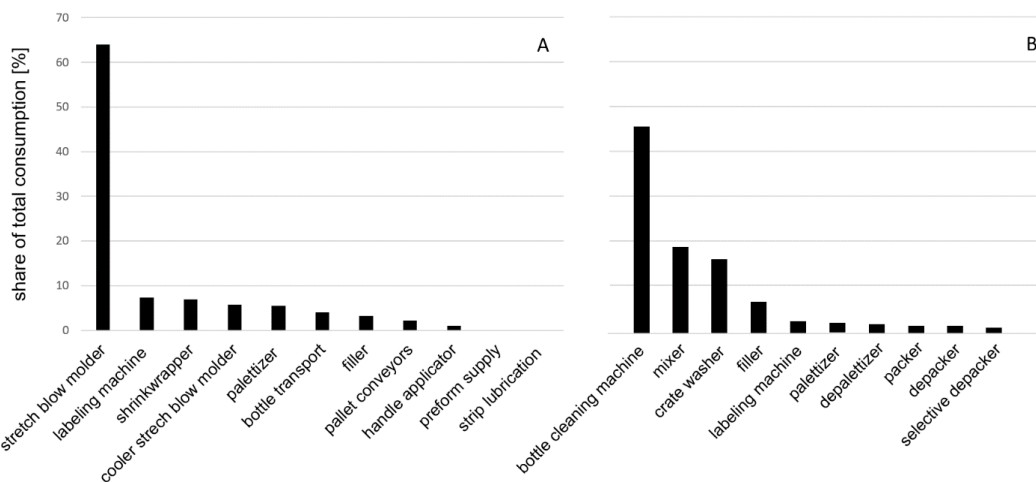

**Figure 6.** Consumption structure of a PET bottling plant (**A**) and a glass bottling plant (**B**). Own measurements for *n* = 1 production line of each type, showing the main consumers: stretch-blow molders and bottle-cleaning machines. The average consumption of the stretch-blow molder is 157 kW, whereas that of the bottle-cleaning machine is 49 kW.

It was found that there is a correlation between the consumption behavior (see, e.g., Table 7) and the occurring operational states. A simple consideration of the specified consumption values will not result in an accurate forecast, as the measurements have a proven deviation from the actual consumption value. The availability of the machine, defined as the share of production time to the total time, and the resulting differing consumption during the occurring operational states will influence the specific energy consumption. Therefore, in the following chapter, the operational-state-related energy consumption behavior will be analyzed in order to provide a reliable analysis of the influence of the operational state and the detailed characterization of the consumption behavior.

**Table 7.** Analysis of the consumption peaks of the considered machines.

| Plant | Machine | Number of Main Peaks | Correlation to the Operational State |
|---|---|---|---|
| Bottling plant 1 | Depalletizer | 3 | (1) Related to inactive times<br>(2) Related to downtime, multimodal peak<br>(3) Extended peak, related to running time |
| | Unpacker | 3 | (1) Related to inactive times<br>(2) Related to downtime, multimodal peak<br>(3) Extended multimodal peak, related to running time, not clearly defined (see the example in Figures 9 and 10) |
| | Selective depacker | 3 | (1) Related to inactive times<br>(2) Multimodal peak, related to downtime (highest peak due to extended downtimes)<br>(3) Related to running time, extended peak |
| | Bottle-cleaning machine | 3 | (1) Related to inactive times<br>(2) Minor peak related to heating-up processes, main peak related to running time<br>(3) Related to operation |
| | Filler | 4 | (1) Multimodal peak, related to inactive times<br>(2) Related to downtime, multimodal peak<br>(3) Related to running time<br>*Note*. The speed of the machine was reduced, resulting in a reduced energy consumption for >1 min after the beginning of a downtime due to technological reasons (emptying of the machine). This results in a multimodal peak (2) |

**Table 7.** *Cont.*

| Plant | Machine | Number of Main Peaks | Correlation to the Operational State |
|---|---|---|---|
| | Labeler | 3 | (1) Multimodal peak, related to inactive times<br>(2) Multimodal peak, related to downtime<br>(3) Related to running time, extended peak |
| | Packer | 3 | (1) Related to inactive times<br>(2) Related to downtime<br>(3) Related to running time, extended peak |
| | Mixer | 3 | No direct correlation found |
| | Palletizer | 3 | (1) Related to inactive times<br>(2) Related to downtime<br>(3) Related to running time, extended peak |
| | Crate washer | 2 | (1) Related to inactive times/downtime<br>(2) Related to running time |
| Bottling plant 2 | Stretch-blow molder | 3 | (1) Related to inactive times<br>(2) Related to downtime<br>(3) Related to running time |
| | Labeler | 3 | (1) Related to inactive times<br>(2) Related to downtime<br>(3) Related to running time |
| | Shrink packer | 4 | (1) Related to inactive times<br>(2) Related to downtime<br>(3) Related to downtime<br>(4) Related to running time |
| | Packer | 3 | (1) Related to inactive times<br>(2) Related to downtimes<br>(3) Related to running time |
| | Filler | 4 | (1) Related to inactive times, multimodal peak<br>(2) Related to downtime, multimodal, not clearly defined<br>(3) (a, b, c) Three main peaks all related to running time, different products with different machine speeds produced on the machine during the measurement time. |
| | Handle application | 3 | (1) Related to inactive times<br>(2) Related to downtime, multimodal peak<br>(3) Related to running time |
| Bottling plant 3 | Bottle-cleaning machine | 3 | (1) Related to inactive times<br>(2) Minor peak related to heating-up processes, major peak related to downtime<br>(3) Related to running time |
| Bottling plant 4 | Bottle-cleaning machine | 3 | (1) Related to inactive times<br>(2) Minor peak related to heating-up processes, major peak related to downtime<br>(3) Related to running time |
| Bottling plant 5 | Stretch-blow molder | 2 | (1) Related to downtime<br>(2) Related to running time<br>*Note.* The measurement time was only 8 h; no inactive times were considered. |
| Packaging machine A | | 2 | (1) Related to inactive times, related to downtime<br>(2) Related to running time<br>*Note.* The measurement time was only 8 h; no inactive times were considered. There was no significant change in consumption owing to the variation of materials/production parameters. |

The energy consumption of the conveyor elements was measured to be 6.2% of the total consumption for the PET bottling plant and should ideally be considered in future measurements. Owing to the large number of single drives, the measurement requires some effort. If possible, the measurement can be summarized to one or two areas. Because of the limited number of devices for measurement, no values are available for the returnable glass bottling plant in this work.

### 3.3. Operational-State-Related Energy Consumption Behavior

#### 3.3.1. Analysis of Empirical Energy Consumption Data in Correlation to Operational State Data

For all machines, the frequency distributions of the measured discrete effective electrical power values (kW), plotted as histograms, were analyzed. The following figures show an example of a bottle-cleaning machine, which is one of the main consumers in the bottling process for returnable glass bottles according to the measurement shown in Figure 6.

Figure 7 shows the measured discrete effective electrical power values, plotted as a histogram, for 42 days of measurement. Three main peaks were identified. Peak 1 indicates low consumption, which is significantly larger than Peaks 2 and 3. This is the result of the longer idle times (weekends, nights) during the measurement. Peak 2 shows reduced consumption and Peak 3 shows the highest consumption in kilowatt. The peaks are not all clearly delineated, but they are merged into each other (Peaks 2 and 3).

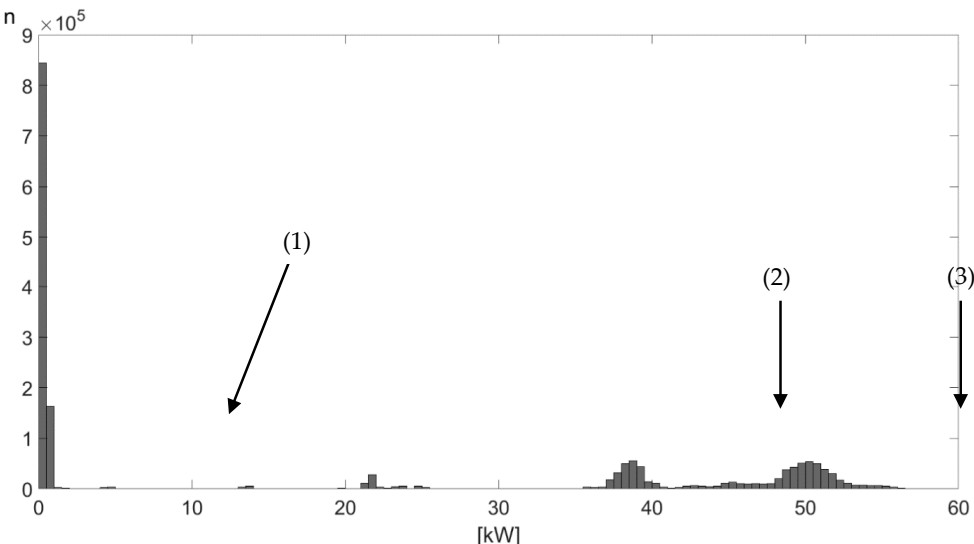

**Figure 7.** Histogram of the measured effective electrical power data of bottle-cleaning machine A over a period of 42 days: three main peaks (Peaks 1, 2, and 3).

While comparing the consumption patterns of the machines shown in the 2D line plots of the effective power data with colored state information in the background and a detailed frequency distribution histogram (right part of Figure 8), a correlation between the identified peaks and the operational states can be proven. Figure 8 shows the details of the 2D line plot mapping the electrical energy consumption pattern of production time interrupted by a tailback, lack, or equipment failure situation for a bottle-cleaning machine (Machine A). The effective electrical power value changes with the operational state changes of the machine.

The two peaks in the histogram (Figures 7 and 8, right) fit the value of the almost constant performance level that the machine reaches after a state change. Different types of peaks related to the running times (see the time model in Figure 1) were found for the machines. The peaks of the machines with a steady operational behavior (bottle filling machines, bottle-cleaning machines, thermoforming machines, etc.) were high and narrow,

whereas they were flat and extended for cycling and intermittently operating machines (packer, palletizer). Figure 9 shows this on the example of a frequency distribution histogram for an unpacking machine. Figure 10 shows the correlation between the histograms and 2D line plots for the unpacker. The alternating values for the effective electrical power during the operational state resulted in a flat and extended peak.

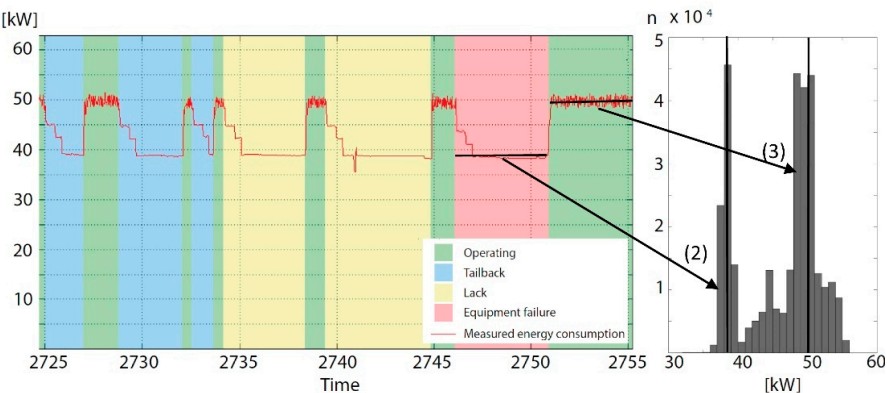

**Figure 8.** Bottle-cleaning machine A. Details of a typical consumption pattern and related operational states (**left**) and a histogram of the measured values for the operating time (here: a time period of 14 days, **right** in the figure), adapted from [32], Isabel Osterroth, 2017.

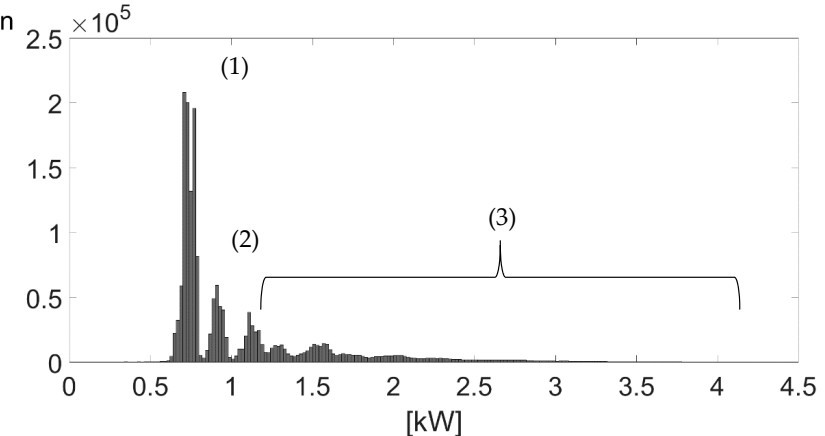

**Figure 9.** Histogram of the effective electrical power data of the intermittently operating machine (unpacker) over a period of 14 days: three main peaks with one flat and extended peak (Peak 3) between 1 and 4 kW.

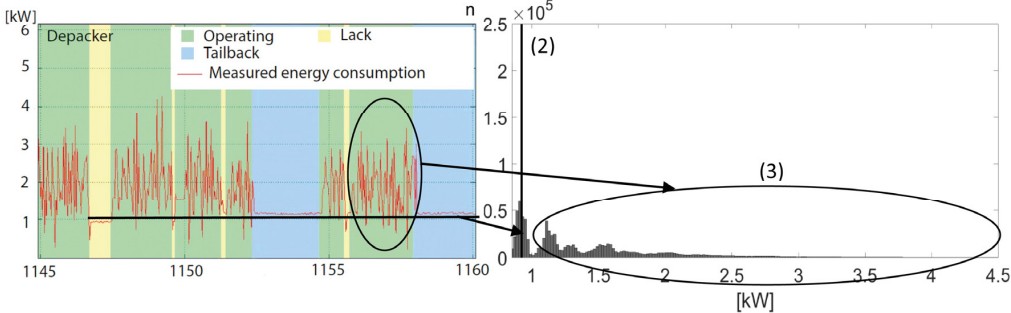

**Figure 10.** Unpacker A. Details of a typical consumption pattern and related operational states (**left**), and a histogram of the measured values for the operating time (**right**).

"Lack", "tailback", "equipment failure", and "held" show similar values in the consumption pattern. This could be verified in a more detailed analysis of the consumption

behavior during the single states. Figure 11 shows the average values of the effective power of single states for an example time period of 4 h. No significant difference between the total consumption values of "lack" and "tailback" was found. The occurring equipment failures were shorter than any lack and tailback situation in this time period. As illustrated in Figure 8, the value of effective power decreases depending on the time of the chance until a constant value is reached. Owing to the short duration of the state equipment failure, the lower constant value of effective power was not reached. This resulted in a higher average effective power value. Any differences were due to the fact that equipment failure is shorter than lack/tailback events.

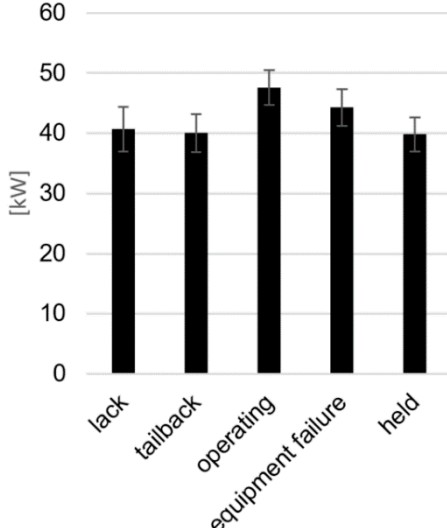

**Figure 11.** Average effective power (kW) and standard deviation (kW) for occurring operational states for bottle-cleaning machine A over an example time period of t = 4 h.

For all other investigated machines, except for the mixer, clearly definable peaks were found. The number of peaks identified and the description of the peaks for all machines are summarized in Table 7. A minimum of two peaks were found for all machines. For most machines, three main peaks were found.

### 3.3.2. Consumption Behavior during State Transitions

Evaluating the frequency distributions of the packaging and bottling machines, it should be noted that the observed peaks are not clearly distinguished, but rather they merge into each other. The measured data shows some machine state changes taking place immediately (e.g., packer). On the other hand, in fillers or bottle-cleaning machines, the states change stepwise by state transitions according to the machine's function (e.g., bringing the product out of the machine, stepwise switching of the pumps for technological reasons; see Figure 12). State transitions are reproducible, as shown in the example in Figure 12 for two machines and 20 state transitions (change from operating to equipment failure). No significant correlation was found between the duration of a downtime (0–180 s) and the energy consumption within the first 60 s of the following running time (see Figure 13). The measurement methods have been shown to be unsuitable for short-term peak loads resulting from the powering up.

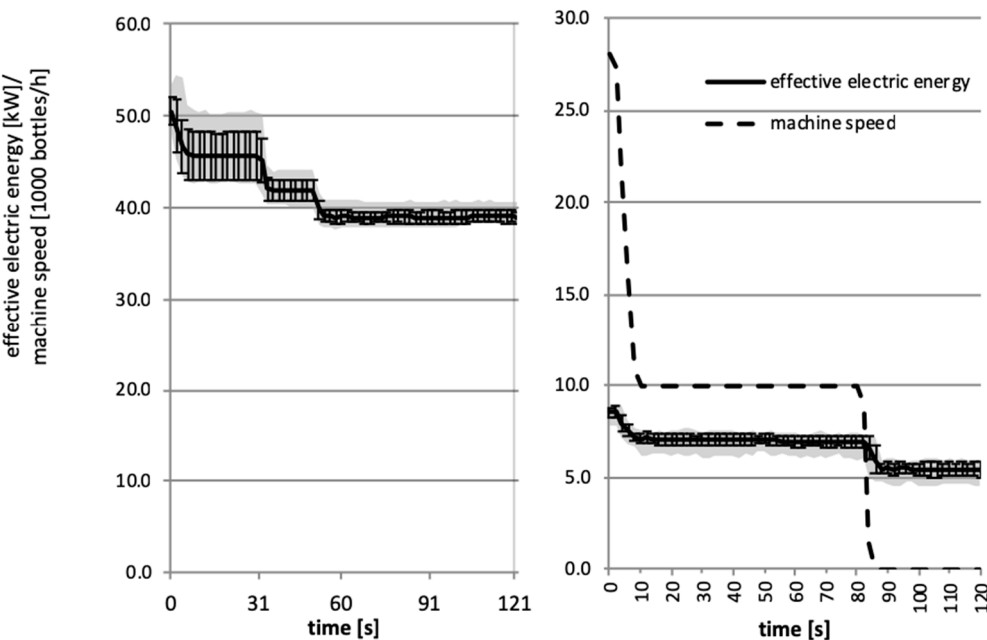

**Figure 12.** Reproducible state transition from operating to equipment failure for *n* = 20 chances: average of the effective electrical energy, standard deviation, and maxima and minima (gray) of the measured values for a bottle filling machine (**right**) and a bottle-cleaning machine (**left**).

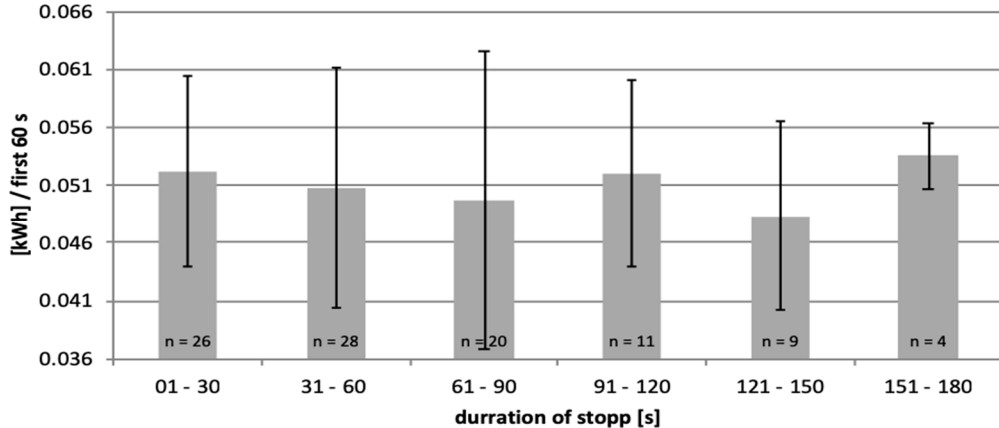

**Figure 13.** Average effective consumption (kWh) over the first 60 s of production, as well as the standard deviation (kWh) sorted by the duration of the downtime before the measurement of a labeling machine. No significant correlation between the duration of the stop and the energy consumption within the first 60 s of production was found.

Figure 14 shows the influence of the machine speed on the energy consumption. While the machine speed was reduced by more than 60% in the first step of state transition, the energy consumption only reduced by 17.5%. In the second step, the machine speed was further reduced to zero, but still more than 60% of the initial energy was consumed by the machine. The influence of the machine speed on the electrical energy consumption is machine-specific, as it depends on the type of consumer within the machine (e.g., electrical drives, pumps, or compressed air), which is not necessarily correlated to the machine speed.

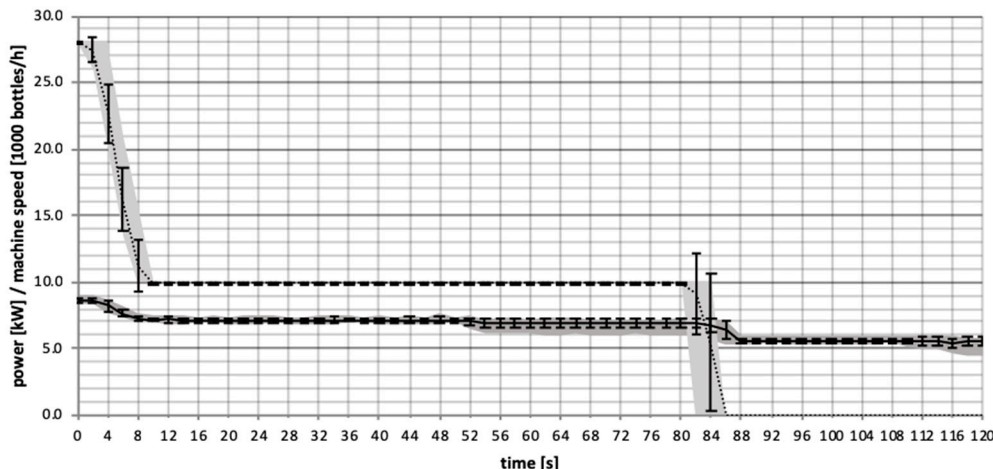

**Figure 14.** Bottle filling machine 1. Reproducible machine speed and influence on the consumption behavior after state transitions from production to equipment failure (*n* = 20 state transitions). A machine speed reduction of −64% results in a reduction of effective electrical power of −17.5%.

### 3.3.3. Discussion of the Operational-State-Related Energy Consumption Behavior

The measured effective electrical energy data proves a correlation between the consumption behavior and the machine's operating state. An operational-state-related approach to modeling suits all machines well, except for mixers. A mixer, showing defined peaks in the frequency distribution not directly matching the operational states, is a process machine influenced by the upstream production process and cannot be described by the same models as processing packaging and bottling machines. All other machines analyzed showcased a direct number of peaks in the frequency distribution of the measured consumption values, as shown, e.g., in Figures 7 and 9 and summarized in Table 7. Correlating the frequency distributions with the colored 2D line plots showed an energy demand level matching the changing operational states. Some machines show multimodal peaks, which can be related to single states (mainly to downtimes). The multimodal frequency distribution peak indicates a machine-specific shutdown behavior. In order to create a detailed description of the machine consumption behavior, state changes in accordance with the elapsed time can be utilized. State changes take place as state transitions, as shown in Figures 12 and 13. They can be described as reproducible based on the analyzed data. The time after a state change also influences the level of effective electrical power and can be used to describe the behavior during state changes in models. For extended modeling (e.g., for simulation cases), the duration of transitions can be determined by data analysis or from the PLC.

The measurements show the operational state being a major influence on the energy consumption of packaging and bottling machines in contrast to the machine speed. While machine speed is directly correlated to the operational state but seems to be not suitable for the modeling of the consumption behavior (shown in Figure 14), the operational state related consumption levels are assumed as suitable to describe the energy consumption behavior of bottling plants. The number of peaks was found as reproducible, regardless of what product was produced; however, the position of the peaks on the x-axis shifts (average consumption in kilowatt). Therefore, the product and process conditions do not influence the state dependency in general, but only the value of the energy level. In order to characterize the energy consumption behavior, the system boundaries, including the nominal speed of the machine, have to be defined, as well as the reference product and process.

## 4. Summary and Outlook

PET and returnable glass bottling plants both have a main consumer significantly influencing the total consumption (bottle-cleaning machines and stretch-blow molders).

Based on the results, it can be said that the energy consumption of food-packaging and bottling machines is described by a limited discrete number of energy states related to the operational state. The product type produced by the machine might influence the discrete value of the described energy states and can be considered as a parameter in modelling approaches. The observed energy states can be mapped to common models describing the operational state behavior and are comparable for all the considered machines. State changes can take place immediately or in time-dependent state transitions, resulting in a constant value after a certain time. The machine speed is not directly correlated to the energy consumption and therefore is not suitable for the energy modeling of the machines. The basic concept of operational-state-related energy consumption can be used for future model-based forecasting of the electrical energy consumption of food-packaging and beverage-bottling machines. The consumption values during non-productive times are still high and should be analyzed in more detail for optimization purposes in future research. Potential technical and technological changes, as well as changes in the plant automation, might lead to reduced energy demand during down times, supporting the ambitious industry targets on sustainability. For a state-based modeling and simulation approach, it is assumed that, for most machines, a model can be simplified to three main consumption states (inactive, standby, and production). The described energy demand level can be regarded as nearly constant. Furthermore, a state-related model can be used for cause-related energy analyses (e.g., to find energy optimization potentials related to downtimes) and future specifications of machines (e.g., decision criteria for improved total cost of ownership considerations). Future research can focus on operational-state-related modeling and forecasting in order to develop complex optimization strategies taking into account both the operational state behavior of the interlinked plants and the specific energy behavior of the single machines in the system.

**Author Contributions:** Conceptualization, I.A.O. and T.V.; methodology, I.A.O.; formal analysis, I.A.O.; investigation, I.A.O.; data curation, I.A.O.; validation, I.A.O.; writing—original draft preparation, I.A.O.; writing—review and editing, T.V.; visualization, I.A.O.; supervision, T.V.; project administration, I.A.O. and T.V. All authors have read and agreed to the published version of the manuscript.

**Funding:** This research received no external funding.

**Institutional Review Board Statement:** Not applicable.

**Informed Consent Statement:** Not applicable.

**Conflicts of Interest:** The authors declare no conflict of interest.

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
