# Peer review of "Energy Consumption of Beverage-Bottling Machines"

_sustainability, doi:10.3390/su13179880_

Round 1

Reviewer 1 Report

The manuscript entitled "Energy consumption of beverage-bottling machines" describes the energy consumption in bottling machines. The manuscript's structure seems to be adopted from a PG thesis; hence, a substantial revision is needed to demonstrate a good-flow reading to readers.

Although the introduction is too long, some statements need to be addressed appropriately.

Section 2 should be shifted to the Introduction.

Also, there is no consistency between different parts of the manuscript. Looking at the abstract and conclusion sections shows this matter as well.     

Author Response

Dear Reviewer,

thank you very much for your review and the valuable comments to improve our submission. Please find my comments below:

Thank you for your suggestions to improve the structure of the paper. I’ve included Section 2 in the introduction and reworked the structure to provide a better flow for the reader and more cosistency. The new structure of the Introduction is now the following:

  • General introduction, adressing the need for analysis, suitable metrics and optimization tools
  • Overview of energy consumption in bottling, introducing bottling plants and summerizing the state-of-the-art of energy performance indicators in bottling
  • Overview of existing state-based models in scientfic literature
  • Purpose of the paper

I’ve reworked the main statements of the introduction and the conclusion to be more precise and to improve the consistency.

Kind regards

Isabel Osterroth 

Reviewer 2 Report

I read the paper with great interest. It addresses an important, though relatively niche aspect of sustainability research.

In this, it entails an important contribution to science by performing a state-of-the-art analysis of energy consumption and efficiency on the machine level. The employed quantitative approach is sound and research is methodologically well designed.

If anything, it could benefit from a few more references of state-of-the-art research articles from top journals.

Potential for future research and potential for industry application of findings could be discussed slightly more broadly.

All in all, I believe the paper to be appropriate for publication in this journal with above minor suggestions for improvement.

Author Response

Dear Reviewer,

thank you very much for your positive feedback on our submission. We highly appreciate your suggestions to further improve the submission. Please find attached my comments below:

If anything, it could benefit from a few more references of state-of-the-art research articles from top journals.

I’ve included two more references, e.g. by from the Journal of Cleaner Production to improve the quality of the submission.

Potential for future research and potential for industry application of findings could be discussed slightly more broadly.

Thank you, I’ve modified the Summary and Outlook section, to be more specific on the potential for future research and industrial application.

Kind regards

Isabel Osterroth

Round 2

Reviewer 1 Report

The comments have been addressed in the revised version.